# ReMixMatch: Semi-Supervised Learning with Distribution Alignment and Augmentation Anchoring

**David Berthelot, Nicholas Carlini, Ekin D. Cubuk, Alex Kurakin, Han Zhang, Colin Raffel**
Google Research
`{dberth,ncarlini,cubuk,kurakin,zhanghan,craffel}@google.com`

**Kihyuk Sohn**
Google Cloud AI
`kihyuks@google.com`

## ABSTRACT

We improve the recently-proposed "MixMatch" semi-supervised learning algorithm by introducing two new techniques: distribution alignment and augmentation anchoring. *Distribution alignment* encourages the marginal distribution of predictions on unlabeled data to be close to the marginal distribution of ground-truth labels. *Augmentation anchoring* feeds multiple strongly augmented versions of an input into the model and encourages each output to be close to the prediction for a weakly-augmented version of the same input. To produce strong augmentations, we propose a variant of AutoAugment which learns the augmentation policy while the model is being trained. Our new algorithm, dubbed ReMixMatch, is significantly more data-efficient than prior work, requiring between $5\times$ and $16\times$ less data to reach the same accuracy. For example, on CIFAR-10 with 250 labeled examples we reach $93.73\%$ accuracy (compared to MixMatch's accuracy of $93.58\%$ with $4,000$ examples) and a median accuracy of $84.92\%$ with just **four** labels per class. We make our code and data open-source at `https://github.com/google-research/remixmatch`.

## 1 INTRODUCTION

Semi-supervised learning (SSL) provides a means of leveraging unlabeled data to improve a model's performance when only limited labeled data is available. This can enable the use of large, powerful models when labeling data is expensive or inconvenient. Research on SSL has produced a diverse collection of approaches, including consistency regularization (Sajjadi et al., 2016; Laine & Aila, 2017) which encourages a model to produce the same prediction when the input is perturbed and entropy minimization (Grandvalet & Bengio, 2005) which encourages the model to output high-confidence predictions. The recently proposed "MixMatch" algorithm (Berthelot et al., 2019) combines these techniques in a unified loss function and achieves strong performance on a variety of image classification benchmarks. In this paper, we propose two improvements which can be readily integrated into MixMatch's framework.

First, we introduce "distribution alignment", which encourages the distribution of a model's aggregated class predictions to match the marginal distribution of ground-truth class labels. This concept was introduced as a "fair" objective by Bridle et al. (1992), where a related loss term was shown to arise from the maximization of mutual information between model inputs and outputs. After reviewing this theoretical framework, we show how distribution alignment can be straightforwardly added to MixMatch by modifying the "guessed labels" using a running average of model predictions.

Second, we introduce "augmentation anchoring", which replaces the consistency regularization component of MixMatch. For each given unlabeled input, augmentation anchoring first generates a weakly augmented version (e.g. using only a flip and a crop) and then generates multiple strongly augmented versions. The model's prediction for the weakly-augmented input is treated as the basis

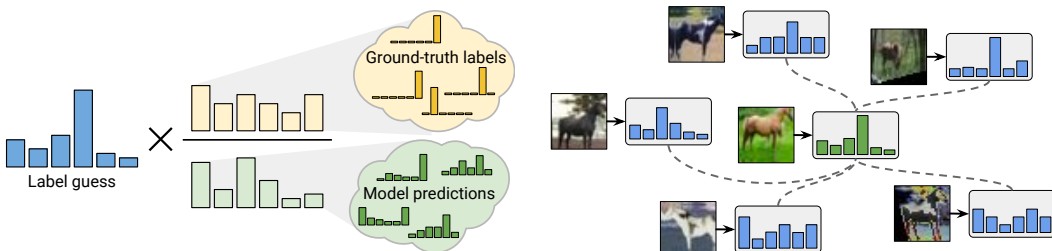

Figure 1: Distribution alignment. Guessed label distributions are adjusted according to the ratio of the empirical ground-truth class distribution divided by the average model predictions on unlabeled data.

Figure 2: Augmentation anchoring. We use the prediction for a weakly augmented image (green, middle) as the target for predictions on strong augmentations of the same image (blue).

of the guessed label for all of the strongly augmented versions. To generate strong augmentations, we introduce a variant of AutoAugment (Cubuk et al., 2018) based on control theory which we dub "CTAugment". Unlike AutoAugment, CTAugment learns an augmentation policy alongside model training, making it particularly convenient in SSL settings.

We call our improved algorithm "ReMixMatch" and experimentally validate it on a suite of standard SSL image benchmarks. ReMixMatch achieves state-of-the-art accuracy across all labeled data amounts, for example achieving an accuracy of $93.73\%$ with 250 labels on CIFAR-10 compared to the previous state-of-the-art of $88.92\%$ (and compared to $96.09\%$ for fully-supervised classification with 50,000 labels). We also push the limited-data setting further than ever before, ultimately achieving a median of $84.92\%$ accuracy with only 40 labels (just 4 labels per class) on CIFAR-10. To quantify the impact of our proposed improvements, we carry out an extensive ablation study to measure the impact of our improvements to MixMatch. Finally, we release all of our models and code to facilitate future work on semi-supervised learning.

## 2 BACKGROUND

The goal of a semi-supervised learning algorithm is to learn from unlabeled data in a way that improves performance on labeled data. Typical ways of achieving this include training against "guessed" labels for unlabeled data or optimizing a heuristically-motivated objective that does not rely on labels. This section reviews the semi-supervised learning methods relevant to ReMixMatch, with a particular focus on the components of the MixMatch algorithm upon which we base our work.

**Consistency Regularization**   Many SSL methods rely on consistency regularization to enforce that the model output remains unchanged when the input is perturbed. First proposed in (Bachman et al., 2014), (Sajjadi et al., 2016) and (Laine & Aila, 2017), this approach was referred to as "Regularization With Stochastic Transformations and Perturbations" and the "Π-Model" respectively. While some work perturbs adversarially (Miyato et al., 2018) or using dropout (Laine & Aila, 2017; Tarvainen & Valpola, 2017), the most common perturbation is to apply domain-specific data augmentation (Laine & Aila, 2017; Sajjadi et al., 2016; Berthelot et al., 2019; Xie et al., 2019). The loss function used to measure consistency is typically either the mean-squared error (Laine & Aila, 2017; Tarvainen & Valpola, 2017; Sajjadi et al., 2016) or cross-entropy (Miyato et al., 2018; Xie et al., 2019) between the model's output for a perturbed and non-perturbed input.

**Entropy Minimization**   Grandvalet & Bengio (2005) argues that unlabeled data should be used to ensure that classes are well-separated. This can be achieved by encouraging the model's output distribution to have low entropy (i.e., to make "high-confidence" predictions) on unlabeled data. For example, one can explicitly add a loss term to minimize the entropy of the model's predicted class distribution on unlabeled data (Grandvalet & Bengio, 2005; Miyato et al., 2018). Related to this idea are "self-training" methods (McLachlan, 1975; Rosenberg et al., 2005) such as Pseudo-Label (Lee, 2013) that use the predicted class on an unlabeled input as a hard target for the same input, which implicitly minimizes the entropy of the prediction.

**Standard Regularization**   Outside of the setting of SSL, it is often useful to regularize models in the over-parameterized regime. This regularization can often be applied both when training on labeled and unlabeled data. For example, standard "weight decay" (Hinton & van Camp, 1993) where the $L^2$ norm of parameters is minimized is often used alongside SSL techniques. Similarly, powerful MixUp regularization (Zhang et al., 2017) which trains a model on linear interpolants of inputs and labels has recently been applied to SSL (Berthelot et al., 2019; Verma et al., 2019).

**Other Approaches**   The three aforementioned categories of SSL techniques does not cover the full literature on semi-supervised learning. For example, there is a significant body of research on "transductive" or graph-based semi-supervised learning techniques which leverage the idea that unlabeled datapoints should be assigned the label of a labeled datapoint if they are sufficiently similar (Gammerman et al., 1998; Joachims, 2003; 1999; Bengio et al., 2006; Liu et al., 2018). Since our work does not involve these (or other) approaches to SSL, we will not discuss them further. A more substantial overview of SSL methods is available in (Chapelle et al., 2006).

## 2.1   MixMatch

MixMatch (Berthelot et al., 2019) unifies several of the previously mentioned SSL techniques. The algorithm works by generating "guessed labels" for each unlabeled example, and then using fully-supervised techniques to train on the original labeled data along with the guessed labels for the unlabeled data. This section reviews the necessary details of MixMatch; see (Berthelot et al., 2019) for a full definition.

Let $\mathcal{X} = \big\{(x_b, p_b) : b \in (1, \dots, B)\big\}$ be a batch of labeled data and their corresponding one-hot labels representing one of $L$ classes and let $\hat{x}_b$ be augmented versions of these labeled examples. Similarly, let $\mathcal{U} = \big\{u_b : b \in (1, \dots, B)\big\}$ be a batch of unlabeled examples. Finally, let $p_{\text{model}}(y \mid x; \theta)$ be the predicted class distribution produced by the model for input $x$.

MixMatch first produces $K$ weakly augmented versions of each unlabeled datapoint $\hat{u}_{b,k}$ for $k \in \{1, \dots, K\}$. Then, it generates a "guessed label" $q_b$ for each $u_b$ by computing the average prediction $\bar{q}_b$ across the $K$ augmented versions: $\bar{q}_b = \frac{1}{K} \sum_k p_{\text{model}}(y \mid \hat{u}_{b,k}; \theta)$. The guessed label distribution is then sharpened by adjusting its temperature (i.e. raising all probabilities to a power of $1/\text{T}$ and renormalizing). Finally, pairs of examples $(x_1, p_1), (x_2, p_2)$ from the combined set of labeled examples and unlabeled examples with label guesses are fed into the MixUp (Zhang et al., 2017) algorithm to compute examples $(x', p')$ where $x' = \lambda x_1 + (1 - \lambda) x_2$ for $\lambda \sim \text{Beta}(\alpha, \alpha)$, and similarly for $p'$. Given these mixed-up examples, MixMatch performs standard fully-supervised training with minor modifications. A standard cross-entropy loss is used for labeled data, whereas the loss for unlabeled data is computed using a mean square error (i.e. the Brier score (Brier, 1950)) and is weighted with a hyperparameter $\lambda_{\mathcal{U}}$. The terms $K$ (number of augmentations), $T$ (sharpening temperature), $\alpha$ (MixUp $\text{Beta}$ parameter), and $\lambda_{\mathcal{U}}$ (unlabeled loss weight) are MixMatch's hyperparameters. For augmentation, shifting and flipping was used for the CIFAR-10, CIFAR-100, and STL-10 datasets, and shifting alone was used for SVHN.

## 3   ReMixMatch

Having introduced MixMatch, we now turn to the two improvements we propose in this paper: Distribution alignment and augmentation anchoring. For clarity, we describe how we integrate them into the base MixMatch algorithm; the full algorithm for ReMixMatch is shown in algorithm 1.

### 3.1   Distribution Alignment

Our first contribution is *distribution alignment*, which enforces that the aggregate of predictions on unlabeled data matches the distribution of the provided labeled data. This general idea was first introduced over 25 years ago (Bridle et al., 1992), but to the best of our knowledge is not used in modern SSL techniques. A schematic of distribution alignment can be seen in fig. 1. After reviewing and extending the theory, we describe how it can be straightforwardly included in ReMixMatch.

---

**Algorithm 1** ReMixMatch algorithm for producing a collection of processed labeled examples and processed unlabeled examples with label guesses (cf. Berthelot et al. (2019) Algorithm 1.)

---

1: **Input:** Batch of labeled examples and their one-hot labels $\mathcal{X} = \big\{(x_b, p_b) : b \in (1, \ldots, B)\big\}$, batch of unlabeled examples $\mathcal{U} = \big\{u_b : b \in (1, \ldots, B)\big\}$, sharpening temperature $T$, number of augmentations $K$, Beta distribution parameter $\alpha$ for MixUp.
2: **for** $b = 1$ **to** $B$ **do**
3:     $\hat{x}_b = \text{StrongAugment}(x_b)$    *// Apply strong data augmentation to $x_b$*
4:     $\hat{u}_{b,k} = \text{StrongAugment}(u_b); k \in \{1, \ldots, K\}$    *// Apply strong data augmentation $K$ times to $u_b$*
5:     $\tilde{u}_b = \text{WeakAugment}(u_b)$    *// Apply weak data augmentation to $u_b$*
6:     $q_b = p_{\text{model}}(y \mid \tilde{u}_b; \theta)$    *// Compute prediction for weak augmentation of $u_b$*
7:     $q_b = \text{Normalize}(q_b \times p(y)/\tilde{p}(y))$    *// Apply distribution alignment*
8:     $q_b = \text{Normalize}\big(q_b^{1/T}\big)$    *// Apply temperature sharpening to label guess*
9: **end for**
10: $\hat{\mathcal{X}} = \big((\hat{x}_b, p_b); b \in (1, \ldots, B)\big)$    *// Augmented labeled examples and their labels*
11: $\hat{\mathcal{U}}_1 = \big((\hat{u}_{b,1}, q_b); b \in (1, \ldots, B)\big)$    *// First strongly augmented unlabeled example and guessed label*
12: $\hat{\mathcal{U}} = \big((\hat{u}_{b,k}, q_b); b \in (1, \ldots, B), k \in (1, \ldots, K)\big)$    *// All strongly augmented unlabeled examples*
13: $\hat{\mathcal{U}} = \hat{\mathcal{U}} \cup \big((\tilde{u}_b, q_b); b \in (1, \ldots, B)\big)$    *// Add weakly augmented unlabeled examples*
14: $\mathcal{W} = \text{Shuffle}\big(\text{Concat}(\hat{\mathcal{X}}, \hat{\mathcal{U}})\big)$    *// Combine and shuffle labeled and unlabeled data*
15: $\mathcal{X}' = \big(\text{MixUp}(\hat{\mathcal{X}}_i, \mathcal{W}_i); i \in (1, \ldots, |\hat{\mathcal{X}}|)\big)$    *// Apply MixUp to labeled data and entries from $\mathcal{W}$*
16: $\mathcal{U}' = \big(\text{MixUp}(\hat{\mathcal{U}}_i, \mathcal{W}_{i+|\hat{\mathcal{X}}|}); i \in (1, \ldots, |\hat{\mathcal{U}}|)\big)$    *// Apply MixUp to unlabeled data and the rest of $\mathcal{W}$*
17: **return** $\mathcal{X}', \mathcal{U}', \hat{\mathcal{U}}_1$

---

### 3.1.1 INPUT-OUTPUT MUTUAL INFORMATION

As previously mentioned, the primary goal of an SSL algorithm is to incorporate unlabeled data in a way which improves a model's performance. One way to formalize this intuition, first proposed by Bridle et al. (1992), is to maximize the mutual information between the model's input and output for unlabeled data. Intuitively, a good classifier's prediction should depend as much as possible on the input. Following the analysis from Bridle et al. (1992), we can formalize this objective as

$$\mathcal{I}(y; x) = \iint p(y, x) \log \frac{p(y, x)}{p(y)p(x)} \, \mathrm{d}y \, \mathrm{d}x \tag{1}$$

$$= \mathcal{H}(\mathbb{E}_x[p_{\text{model}}(y|x; \theta)]) - \mathbb{E}_x[\mathcal{H}(p_{\text{model}}(y|x; \theta))] \tag{2}$$

where $\mathcal{H}(\cdot)$ refers to the entropy. See Appendix A for a proof. To interpret this result, observe that the second term in eq. (2) is the familiar entropy minimization objective (Grandvalet & Bengio, 2005), which simply encourages each individual model output to have low entropy (suggesting high confidence in a class label). The first term, however, is not widely used in modern SSL techniques. This term (roughly speaking) encourages that on average, across the entire training set, the model predicts each class with equal frequency. Bridle et al. (1992) refer to this as the model being "fair".

### 3.1.2 DISTRIBUTION ALIGNMENT IN REMIXMATCH

MixMatch already includes a form of entropy minimization via the "sharpening" operation which makes the guessed labels (synthetic targets) for unlabeled data have lower entropy. We are therefore interested in also incorporating a form of "fairness" in ReMixMatch. However, note that the objective $\mathcal{H}(\mathbb{E}_x[p_{\text{model}}(y|x; \theta)])$ on its own essentially implies that the model should predict each class with equal frequency. This is not necessarily a useful objective if the dataset's marginal class distribution $p(y)$ is not uniform. Furthermore, while it would in principle be possible to directly minimize this objective on a per-batch basis, we are instead interested in integrating it into MixMatch in a way which does not introduce an additional loss term or any sensitive hyperparameters.

To address these issues, we incorporate a form of fairness we call "distribution alignment" which proceeds as follows: over the course of training, we maintain a running average of the model's predictions on unlabeled data, which we refer to as $\tilde{p}(y)$. Given the model's prediction $q = p_{\text{model}}(y|u; \theta)$ on an unlabeled example $u$, we scale $q$ by the ratio $p(y)/\tilde{p}(y)$ and then renormalize the result to form a valid probability distribution: $\tilde{q} = \text{Normalize}(q \times p(y)/\tilde{p}(y))$ where

Normalize$(x)_i = x_i / \sum_j x_j$. We then use $\tilde{q}$ as the label guess for $u$, and proceed as usual with sharpening and other processing. In practice, we compute $\tilde{p}(y)$ as the moving average of the model's predictions on unlabeled examples over the last $128$ batches. We also estimate the marginal class distribution $p(y)$ based on the labeled examples seen during training. Note that a better estimate for $p(y)$ could be used if it is known a priori; in this work we do not explore this direction further.

## 3.2 IMPROVED CONSISTENCY REGULARIZATION

Consistency regularization underlies most SSL methods (Miyato et al., 2018; Tarvainen & Valpola, 2017; Berthelot et al., 2019; Xie et al., 2019). For image classification tasks, consistency is typically enforced between two augmented versions of the same unlabeled image. In order to enforce a form of consistency regularization, MixMatch generates $K$ (in practice, $K = 2$) augmentations of each unlabeled example $u$ and averages them together to produce a "guessed label" for $u$.

Recent work (Xie et al., 2019) found that applying stronger forms of augmentation can significantly improve the performance of consistency regularization. In particular, for image classification tasks it was shown that using variants of AutoAugment (Cubuk et al., 2018) produced substantial gains. Since MixMatch uses a simple flip-and-crop augmentation strategy, we were interested to see if replacing the weak augmentation in MixMatch with AutoAugment would improve performance but found that training would not converge. To circumvent this issue, we propose a new method for consistency regularization in MixMatch called "Augmentation Anchoring". The basic idea is to use the model's prediction for a weakly augmented unlabeled image as the guessed label for many strongly augmented versions of the same image.

A further logistical concern with using AutoAugment is that it uses reinforcement learning to learn a policy which requires many trials of supervised model training. This poses issues in the SSL setting where we often have limited labeled data. To address this, we propose a variant of AutoAugment called "CTAugment" which adapts itself online using ideas from control theory without requiring any form of reinforcement learning-based training. We describe Augmentation Anchoring and CTAugment in the following two subsections.

### 3.2.1 AUGMENTATION ANCHORING

We hypothesize the reason MixMatch with AutoAugment is unstable is that MixMatch averages the prediction across $K$ augmentations. Stronger augmentation can result in disparate predictions, so their average may not be a meaningful target. Instead, given an unlabeled input we first generate an "anchor" by applying weak augmentation to it. Then, we generate $K$ strongly-augmented versions of the same unlabeled input using CTAugment (described below). We use the guessed label (after applying distribution alignment and sharpening) as the target for all of the $K$ strongly-augmented versions of the image. This process is visualized in fig. 2.

While experimenting with Augmentation Anchoring, we found it enabled us to replace MixMatch's unlabeled-data mean squared error loss with a standard cross-entropy loss. This maintained stability while also simplifying the implementation. While MixMatch achieved its best performance at only $K = 2$, we found that augmentation anchoring benefited from a larger value of $K = 8$. We compare different values of $K$ in section 4 to measure the gain achieved from additional augmentations.

### 3.2.2 CONTROL THEORY AUGMENT

AutoAugment (Cubuk et al., 2018) is a method for learning a data augmentation policy which results in high validation set accuracy. An augmentation policy consists a sequence of transformation-parameter magnitude tuples to apply to each image. Critically, the AutoAugment policy is learned with supervision: the magnitudes and sequence of transformations are determined via training many models on a proxy task which e.g. involves the use of $4,000$ labels on CIFAR-10 and $1,000$ labels on SVHN (Cubuk et al., 2018). This makes applying AutoAugment methodologically problematic for low-label SSL. To remedy this necessity for training a policy on labeled data, RandAugment (Cubuk et al., 2019) uniformly randomly samples transformations, but requires tuning the hyper-parameters for the random sampling on the validation set, which again is methodologically difficult when only very few (e.g., 40 or 250) labeled examples are available.

Thus, in this work, we develop CTAugment, an alternative approach to designing high-performance augmentation strategies. Like RandAugment, CTAugment also uniformly randomly samples transformations to apply but dynamically infers magnitudes for each transformation during the training process. Since CTAugment does not need to be optimized on a supervised proxy task and has no sensitive hyperparameters, we can directly include it in our semi-supervised models to experiment with more aggressive data augmentation in semi-supervised learning. Intuitively, for each augmentation parameter, CTAugment learns the likelihood that it will produce an image which is classified as the correct label. Using these likelihoods, CTAugment then only samples augmentations that fall within the network tolerance. This process is related to what is called density-matching in Fast AutoAugment (Lim et al., 2019), where policies are optimized so that the density of augmented validation images match the density of images from the training set.

First, CTAugment divides each parameter for each transformation into bins of distortion magnitude as is done in AutoAugment (see Appendix C for a list of the bin ranges). Let $m$ be the vector of bin weights for some distortion parameter for some transformation. At the beginning of training, all magnitude bins are initialized to have a weight set to $1$. These weights are used to determine which magnitude bin to apply to a given image.

At each training step, for each image two transformations are sampled uniformly at random. To augment images for training, for each parameter of these transformations we produce a modified set of bin weights $\hat{m}$ where $\hat{m}_i = m_i$ if $m_i > 0.8$ and $\hat{m}_i = 0$ otherwise, and sample magnitude bins from $\mathrm{Categorical}(\mathrm{Normalize}(\hat{m}))$. To update the weights of the sampled transformations, we first sample a magnitude bin $m_i$ for each transformation parameter uniformly at random. The resulting transformations are applied to a labeled example $x$ with label $p$ to obtain an augmented version $\hat{x}$. Then, we measure the extent to which the model's prediction matches the label as $\omega = 1 - \frac{1}{2L} \sum |p_{\mathrm{model}}(y|\hat{x}; \theta) - p|$. The weight for each sampled magnitude bin is updated as $m_i = \rho m_i + (1 - \rho)\omega$ where $\rho = 0.99$ is a fixed exponential decay hyperparameter.

### 3.3 PUTTING IT ALL TOGETHER

ReMixMatch's algorithm for processing a batch of labeled and unlabeled examples is shown in algorithm 1. The main purpose of this algorithm is to produce the collections $\mathcal{X}'$ and $\mathcal{U}'$, consisting of augmented labeled and unlabeled examples with $\mathrm{MixUp}$ applied. The labels and label guesses in $\mathcal{X}'$ and $\mathcal{U}'$ are fed into standard cross-entropy loss terms against the model's predictions. Algorithm 1 also outputs $\hat{\mathcal{U}}_1$, which consists of a single heavily-augmented version of each unlabeled image and its label guesses *without MixUp applied*. $\hat{\mathcal{U}}_1$ is used in two additional loss terms which provide a mild boost in performance in addition to improved stability:

**Pre-mixup unlabeled loss**  We feed the guessed labels and predictions for example in $\hat{\mathcal{U}}_1$ as-is into a separate cross-entropy loss term.

**Rotation loss**  Recent result have shown that applying ideas from self-supervised learning to SSL can produce strong performance (Gidaris et al., 2018; Zhai et al., 2019). We integrate this idea by rotating each image $u \in \hat{\mathcal{U}}_1$ as $\mathrm{Rotate}(u, r)$ where we sample the rotation angle $r$ uniformly from $r \sim \{0, 90, 180, 270\}$ and then ask the model to predict the rotation amount as a four-class classification problem.

In total, the ReMixMatch loss is

$$\sum_{x,p \in \mathcal{X}'} \mathrm{H}(p, p_{\mathrm{model}}(y|x; \theta)) + \lambda_{\mathcal{U}} \sum_{u,q \in \mathcal{U}'} \mathrm{H}(q, p_{\mathrm{model}}(y|u; \theta)) \tag{3}$$

$$+ \lambda_{\hat{\mathcal{U}}_1} \sum_{u,q \in \hat{\mathcal{U}}_1} \mathrm{H}(q, p_{\mathrm{model}}(y|u; \theta)) + \lambda_r \sum_{u \in \hat{\mathcal{U}}_1} \mathrm{H}(r, p_{\mathrm{model}}(r|\mathrm{Rotate}(u, r); \theta)) \tag{4}$$

**Hyperparameters**  ReMixMatch introduce two new hyperparameters: the weight on the rotation loss $\lambda_r$ and the weight on the un-augmented example $\lambda_{\hat{\mathcal{U}}_1}$. In practice both are fixed $\lambda_r = \lambda_{\hat{\mathcal{U}}_1} = 0.5$. ReMixMatch also shares many hyperparameters from MixMatch: the weight for the unlabeled loss $\lambda_{\mathcal{U}}$, the sharpening temperature $T$, the MixUp Beta parameter, and the number of augmentations $K$. All experiments (unless otherwise stated) use $T = 0.5$, Beta $= 0.75$, and $\lambda_{\mathcal{U}} = 1.5$. We found

using a larger number of augmentations monotonically increases accuracy, and so set $K = 8$ for all experiments (as running with $K$ augmentations increases computation by a factor of $K$).

We train our models using Adam (Kingma & Ba, 2015) with a fixed learning rate of 0.002 and weight decay (Zhang et al., 2018) with a fixed value of 0.02. We take the final model as an exponential moving average over the trained model weights with a decay of 0.999.

## 4 EXPERIMENTS

We now test the efficacy of ReMixMatch on a set of standard semi-supervised learning benchmarks. Unless otherwise noted, all of the experiments performed in this section use the same codebase and model architecture (a Wide ResNet-28-2 (Zagoruyko & Komodakis, 2016) with 1.5 million parameters, as used in (Oliver et al., 2018)).

### 4.1 REALISTIC SSL SETTING

We follow the Realistic Semi-Supervised Learning (Oliver et al., 2018) recommendations for performing SSL evaluations. In particular, as mentioned above, this means we use the same model and training algorithm in the same codebase for all experiments. We compare against VAT (Miyato et al., 2018) and MeanTeacher (Tarvainen & Valpola, 2017), copying the re-implementations over from the MixMatch codebase (Berthelot et al., 2019).

**Fully supervised baseline** To begin, we train a fully-supervised baseline to measure the highest accuracy we could hope to obtain with our training pipeline. The experiments we perform use the same model and training algorithm, so these baselines are valid for all discussed SSL techniques. On CIFAR-10, we obtain an fully-supervised error rate of $4.25\%$ using weak flip + crop augmentation, which drops to $3.62\%$ using AutoAugment and $3.91\%$ using CTAugment. Similarly, on SVHN we obtain $2.70\%$ error using weak (flip) augmentation and $2.31\%$ and $2.16\%$ using AutoAugment and CTAugment respectively. While AutoAugment performs slightly better on CIFAR-10 and slightly worse on SVHN compared to CTAugment, it is not our intent to design a *better* augmentation strategy; just one that can be used *without a pre-training or tuning of hyper-parameters*.

**CIFAR-10** Our results on CIFAR-10 are shown in table 1, left. ReMixMatch sets the new state-of-the-art for all numbers of labeled examples. Most importantly, ReMixMatch is $16\times$ more data efficient than MixMatch (e.g., at 250 labeled examples ReMixMatch has identical accuracy compared to MixMatch at 4,000).

**SVHN** Results for SVHN are shown in table 1, right. ReMixMatch reaches state-of-the-art at 250 labeled examples, and within the margin of error for state-of-the-art otherwise.

|  | CIFAR-10 | | | SVHN | | |
|---|---|---|---|---|---|---|
| Method | 250 labels | 1000 labels | 4000 labels | 250 labels | 1000 labels | 4000 labels |
| VAT | 36.03±2.82 | 18.64±0.40 | 11.05±0.31 | 8.41±1.01 | 5.98±0.21 | 4.20±0.15 |
| Mean Teacher | 47.32±4.71 | 17.32±4.00 | 10.36±0.25 | 6.45±2.43 | 3.75±0.10 | 3.39±0.11 |
| MixMatch | 11.08±0.87 | 7.75±0.32 | 6.24±0.06 | 3.78±0.26 | 3.27±0.31 | 2.89±0.06 |
| ReMixMatch | 6.27±0.34 | 5.73±0.16 | 5.14±0.04 | 3.10±0.50 | 2.83±0.30 | 2.42±0.09 |
| UDA, reported* | 8.76±0.90 | 5.87±0.13 | 5.29±0.25 | 2.76±0.17 | 2.55±0.09 | 2.47±0.15 |

Table 1: Results on CIFAR-10 and SVHN. * For UDA, due to adaptation difficulties, we report the results from Xie et al. (2019) which are not comparable to our results due to a different network implementation, training procedure, etc. For VAT, Mean Teacher, and MixMatch, we report results using our reimplementation, which makes them directly comparable to ReMixMatch's scores.

### 4.2 STL-10

The STL-10 dataset consists of 5,000 labeled $96 \times 96$ color images drawn from 10 classes and 100,000 unlabeled images drawn from a similar—but not identical—data distribution. The labeled

set is partitioned into ten pre-defined folds of 1,000 images each. For efficiency, we only run our analysis on five of these ten folds. We do not perform evaluation here under the Realistic SSL (Oliver et al., 2018) setting when comparing to non-MixMatch results. Our results are, however, directly comparable to the MixMatch results. Using the same WRN-37-2 network (23.8 million parameters), we reduce the error rate by a factor of two compared to MixMatch.

| Method | Error Rate |
|---|---|
| SWWAE | 25.70 |
| CC-GAN | 22.20 |
| MixMatch | $10.18 \pm 1.46$ |
| ReMixMatch (K=1) | $6.77 \pm 1.66$ |
| ReMixMatch (K=4) | $6.18 \pm 1.24$ |

Table 2: STL-10 error rate using 1000-label splits. SWWAE and CC-GAN results are from (Zhao et al., 2015) and (Denton et al., 2016).

| Ablation | Error Rate | Ablation | Error Rate |
|---|---|---|---|
| ReMixMatch | 5.94 | No rotation loss | 6.08 |
| With K=1 | 7.32 | No pre-mixup loss | 6.66 |
| With K=2 | 6.74 | No dist. alignment | 7.28 |
| With K=4 | 6.21 | L2 unlabeled loss | 17.28 |
| With K=16 | 5.93 | No strong aug. | 12.51 |
| MixMatch | 11.08 | No weak aug. | 29.36 |

Table 3: Ablation study. Error rates are reported on a single 250-label split from CIFAR-10.

### 4.3 Towards few-shot learning

We find that ReMixMatch is able to work in extremely low-label settings. By only changing $\lambda_r$ from 0.5 to 2 we can train CIFAR-10 with just **four** labels per class and SVHN with only 40 labels total. On CIFAR-10 we obtain a median-of-five error rate of $15.08\%$; on SVHN we reach $3.48\%$ error and on SVHN with the "extra" dataset we reach $2.81\%$ error. Full results are given in Appendix B.

### 4.4 Ablation Study

Because we have made several changes to the existing MixMatch algorithm, here we perform an ablation study and remove one component of ReMixMatch at a time to understand from which changes produce the largest accuracy gains. Our ablation results are summarized in Table 3. We find that removing the pre-mixup unlabeled loss, removing distribution alignment, and lowering $K$ all hurt performance by a small amount. Given that distribution alignment improves performance, we were interested to see whether it also had the intended effect of making marginal distribution of model predictions match the ground-truth marginal class distribution. We measure this directly in appendix D. Removing the rotation loss reduces accuracy at 250 labels by only 0.14 percentage points, but we find that in the 40-label setting rotation loss is necessary to prevent collapse. Changing the cross-entropy loss on unlabeled data to an L2 loss as used in MixMatch hurts performance dramatically, as does removing either of the augmentation components. This validates using augmentation anchoring in place of the consistency regularization mechanism of MixMatch.

## 5 Conclusion

Progress on semi-supervised learning over the past year has upended many of the long-held beliefs about classification, namely, that vast quantities of labeled data is necessary. By introducing augmentation anchoring and distribution alignment to MixMatch, we continue this trend: ReMixMatch reduces the quantity of labeled data needed by a large factor compared to prior work (e.g., beating MixMatch at 4000 labeled examples with only 250 on CIFAR-10, and closely approaching MixMatch at 5000 labeled examples with only 1000 on STL-10). In future work, we are interested in pushing the limited data regime further to close the gap between few-shot learning and SSL. We also note that in many real-life scenarios, a dataset begins as unlabeled and is incrementally labeled until satisfactory performance is achieved. Our strong empirical results suggest that it will be possible to achieve gains in this "active learning" setting by using ideas from ReMixMatch. Finally, in this paper we present results on widely-studied image benchmarks for ease of comparison. However, the true power of data-efficient learning will come from applying these techniques to real-world problems where obtaining labeling data is expensive or impractical.

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

## A  PROOF OF EQUATION 2

The proof here follows closely Bridle et al. (1992). We begin with the definition

$$\mathcal{I}(y;x) = \iint p(y,x) \log \frac{p(y,x)}{p(y)p(x)} \, \mathrm{d}y \, \mathrm{d}x \tag{5}$$

Rewriting terms we obtain

$$= \int p(x) \, \mathrm{d}x \int p(y|x) \log \frac{p(y|x)}{p(y)} \, \mathrm{d}y \tag{6}$$

$$= \int p(x) \, \mathrm{d}x \int p(y|x) \log \frac{p(y|x)}{\int p(x)p(y|x) \, \mathrm{d}x} \, \mathrm{d}y \tag{7}$$

Then, rewriting both integrals as expectations we obtain

$$= \mathbb{E}_x \left[ \int p(y|x) \log \frac{p(y|x)}{\mathbb{E}_x[p(y|x)]} \, \mathrm{d}y \right] \tag{8}$$

$$= \mathbb{E}_x \left[ \sum_{i=1}^{L} p(y_i|x) \log \frac{p(y_i|x)}{\mathbb{E}_x[p(y_i|x)]} \right] \tag{9}$$

$$= \mathbb{E}_x \left[ \sum_{i=1}^{L} p(y_i|x) \log p(y_i|x) \right] - \mathbb{E}_x \left[ \sum_{i=1}^{L} p(y_i|x) \log \mathbb{E}_x[p(y_i|x)] \right] \tag{10}$$

$$= \mathbb{E}_x \left[ \sum_{i=1}^{L} p(y_i|x) \log p(y_i|x) \right] - \sum_{i=1}^{L} \mathbb{E}_x[p(y_i|x)] \log \mathbb{E}_x[p(y_i|x)]] \tag{11}$$

$$= \mathcal{H}(\mathbb{E}_x[p_{\mathrm{model}}(y|x;\theta)]) - \mathbb{E}_x[\mathcal{H}(p_{\mathrm{model}}(y|x;\theta))] \tag{12}$$

## B  FULL 40 LABEL RESULTS

We now report the full results for running ReMixMatch with just 40 labeled examples. We sort the table by error rate over five different splits (i.e., 40-label subsets) of the training data. High variance is to be expected when choosing so few labeled examples at random.

| Dataset | Split (ordered by error rate) | | | | |
|---|---|---|---|---|---|
| | 1 | 2 | 3 | 4 | 5 |
| CIFAR-10 | 10.88 | 12.65 | 15.08 | 16.78 | 19.49 |
| SVHN | 3.43 | 3.46 | 3.48 | 4.06 | 12.24 |
| SVHN+extra | 2.59 | 2.71 | 2.81 | 3.50 | 15.14 |

Table 4: Sorted error rate of ReMixMatch with 40 labeled examples.

# C   TRANSFORMATIONS INCLUDED IN CTAUGMENT

| Transformation | Description | Parameter | Range |
|---|---|---|---|
| Autocontrast | Maximizes the image contrast by setting the darkest (lightest) pixel to black (white), and then blends with the original image with blending ratio $\lambda$. | $\lambda$ | [0, 1] |
| Blur | | | |
| Brightness | Adjusts the brightness of the image. $B = 0$ returns a black image, $B = 1$ returns the original image. | $B$ | [0, 1] |
| Color | Adjusts the color balance of the image like in a TV. $C = 0$ returns a black & white image, $C = 1$ returns the original image. | $C$ | [0, 1] |
| Contrast | Controls the contrast of the image. A $C = 0$ returns a gray image, $C = 1$ returns the original image. | $C$ | [0, 1] |
| Cutout | Sets a random square patch of side-length ($L \times$image width) pixels to gray. | $L$ | [0, 0.5] |
| Equalize | Equalizes the image histogram, and then blends with the original image with blending ratio $\lambda$. | $\lambda$ | [0, 1] |
| Invert | Inverts the pixels of the image, and then blends with the original image with blending ratio $\lambda$. | $\lambda$ | [0, 1] |
| Identity | Returns the original image. | | |
| Posterize | Reduces each pixel to $B$ bits. | $B$ | [1, 8] |
| Rescale | Takes a center crop that is of side-length ($L \times$image width), and rescales to the original image size using method $M$. | $L$ | [0.5, 1.0] |
| | | $M$ | see caption |
| Rotate | Rotates the image by $\theta$ degrees. | $\theta$ | [-45, 45] |
| Sharpness | Adjusts the sharpness of the image, where $S = 0$ returns a blurred image, and $S = 1$ returns the original image. | $S$ | [0, 1] |
| Shear_x | Shears the image along the horizontal axis with rate $R$. | $R$ | [-0.3, 0.3] |
| Shear_y | Shears the image along the vertical axis with rate $R$. | $R$ | [-0.3, 0.3] |
| Smooth | Adjusts the smoothness of the image, where $S = 0$ returns a maximally smooth image, and $S = 1$ returns the original image. | $S$ | [0, 1] |
| Solarize | Inverts all pixels above a threshold value of $T$. | $T$ | [0, 1] |
| Translate_x | Translates the image horizontally by ($\lambda \times$image width) pixels. | $\lambda$ | [-0.3, 0.3] |
| Translate_y | Translates the image vertically by ($\lambda \times$image width) pixels. | $\lambda$ | [-0.3, 0.3] |

Table 5: The ranges for all of the listed parameters are discretized into 17 equal bins. The only exception is the $M$ parameter of the Rescale transformation, which takes on one of the following six options: anti-alias, bicubic, bilinear, box, hamming, and nearest.

# D   MEASURING THE EFFECT OF DISTRIBUTION ALIGNMENT

Recall that the goal of distribution alignment is to encourage the marginal distribution of the model's predictions $\tilde{p}(y)$ to match the true marginal class distribution $p(y)$. To measure whether distribution alignment indeed has this effect, we monitored the KL divergence between $\tilde{p}(y)$ and $p(y)$ over the course of training. We show the KL divergence for a training run on CIFAR-10 with 250 labels with

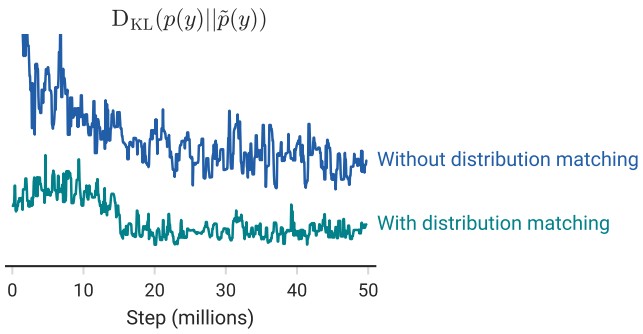

Figure 3: KL divergence between the marginal distribution of model predictions vs. the true marginal distribution of class labels over the course of training with and without distribution alignment. This figure corresponds to a training run on CIFAR-10 with 250 labels.

and without distribution alignment in fig. 3. Indeed, the KL divergence between $\tilde{p}(y)$ and $p(y)$ is significantly smaller throughout training.

## E    CTAUGMENT PARAMETERS EFFECTS

In this section we compare the effects of varying the CTAugment hyper-parameters on CIFAR10 with 250 labels, using the standard ReMixMatch settings. The exponential weight decay $\rho$ does not effect the results significantly while depth and threshold have significant effects. The default settings are highlighted in the table. They appear to perform well and have been shown to be robust across many datasets in our previous experiments.

| Depth | Threshold | $\rho$ | Error rate |
|---|---|---|---|
| 1 | 0.80 | 0.99 | 23.90 |
| 2 | 0.80 | 0.99 | **6.25** |
| 3 | 0.80 | 0.99 | 6.36 |
| 2 | 0.50 | 0.99 | 10.51 |
| 2 | 0.80 | 0.99 | **6.25** |
| 2 | 0.90 | 0.99 | 10.80 |
| 2 | 0.95 | 0.99 | 18.47 |
| 2 | 0.80 | 0.9 | 6.15 |
| 2 | 0.80 | 0.99 | **6.25** |
| 2 | 0.80 | 0.999 | 6.02 |

Table 6: Effects of hyper-parameters for CTAugment, the bold results are the default settings used for all experiments.

