# OpenReview forum: "ReMixMatch: Semi-Supervised Learning with Distribution Matching and Augmentation Anchoring"
_ICLR.cc/2020/Conference — Accept (Poster)_

### Official Review · AnonReviewer2 · 2019-10-22
**Official Blind Review #2**

**Rating:** 6

**Review:**

This paper proposes two modifications for the MixMatch method [1] and achieves improved accuracy on a range of semi-supervised benchmarks. The first modification enforces the distribution of predicted labels to match the distribution of labeled data. The second modification is adding a learned data augmentation strategy, and adapting the method to work with strong data augmentation. The final method is titled ReMixMatch, and improves significantly over MixMatch, especially in low-data regime.

The main contribution of the paper is really strong empirical results. The method achieves state of the art results or close to that on multiple benchmarks, with especially large gains in settings with very scarce labeled data, like 40 labels on CIFAR-10.

Another important contribution is the learned data augmentation strategy, which as far as I understand is novel and overcomes some of the limitations of  existing learned data augmentation techniques. However, the explanation of the strategy wasn’t very clear for me, and the authors didn’t frame it as a major contribution.

The main drawback of the paper is that it seems to be more engineering-focused, and doesn’t provide much insight into semi-supervised learning. The paper can be summarized as adding two modifications to mix-match, and getting better results. The final method becomes fairly involved. Mix-Match is already an elaborate method, and ReMixMatch additionally introduces learned data augmentation, an additional loss term for matching label distributions between labeled and unlabeled data, consistency-loss, and a self-supervised loss (section 3.3).

For the reasons above, I think the paper is borderline, but I am currently voting for acceptance based on the strong empirical performance. At the same time, I think the paper can be made stronger and more interesting to read, if the authors added some experiments aimed at understanding the proposed modifications.

One set of experiments that I think would be interesting is aimed at understanding the distribution-matching part. For example, it would be great if the author could demonstrate that without this loss term the distribution of the predicted classes is wrong in the experiments from Section 4. It would also be interesting to see an experiment where the labeled data has a skewed distribution of classes, but we provide the method with information about the true class distribution, and demonstrating that this information helps predictive performance.

For the learnable data augmentation it would be great if the authors could provide more insight into the method, how it works, and why is it better than the alternatives. Just analyzing the learned data augmentation in different settings and adding more intuition for what happens would make the paper more insightful and interesting to read.

On a more minor note, the paper [1] seems to report 4.95 accuracy for MixMatch on CIFAR-10 with 4k labels, while in this paper it’s being reported as 6.24. What is the reason for the difference? Another paper, [2], reports very competitive results on CIFAR-10 for 4k labels. I would recommend discussing these results briefly in the paper. At the same time the empirical performance of ReMixMatch is really impressive, and I don’t think the results in [1] and [2] affect their significance.

[1] MixMatch: A Holistic Approach to Semi-Supervised Learning
David Berthelot, Nicholas Carlini, Ian Goodfellow, Nicolas Papernot, Avital Oliver, Colin Raffel

[2] There Are Many Consistent Explanations of Unlabeled Data: Why You Should Average
Ben Athiwaratkun, Marc Finzi, Pavel Izmailov, Andrew Gordon Wilson


**Experience Assessment:**

I have published one or two papers in this area.

**Review Assessment: Checking Correctness Of Derivations And Theory:**

N/A

**Review Assessment: Checking Correctness Of Experiments:**

I assessed the sensibility of the experiments.

**Review Assessment: Thoroughness In Paper Reading:**

I read the paper at least twice and used my best judgement in assessing the paper.

---

> ### Author Response · Authors · 2019-11-14
> **Official response to blind review #2**
>
> 1. One set of experiments that I think would be interesting is aimed at understanding the distribution-matching part. For example, it would be great if the author could demonstrate that without this loss term the distribution of the predicted classes is wrong in the experiments from Section 4.
> A: We actually ran this experiment, where we monitored the KL divergence between the marginal distribution of model predictions and the true marginal distribution of labeled data over the course of training (with and without distribution matching). We added the results of this experiment to the appendix of the latest revision.
>
> 2. For the learnable data augmentation it would be great if the authors could provide more insight into the method, how it works, and why is it better than the alternatives.
> A: For space reasons we provided only a short description of CTAugment, and how it differs from AutoAugment. We will include a longer treatment in the appendix.
>
> 3. On a more minor note, the paper [1] seems to report 4.95 accuracy for MixMatch on CIFAR-10 with 4k labels, while in this paper it’s being reported as 6.24. What is the reason for the difference?
> A: The 4.95 error rate in the MixMatch paper is in Table 1 which is the result when using a larger model (26 million parameters). Our results are comparable to the WRN-28-2 results (as used in the "Realistic Evaluation of Semi-Supervised Learning Algorithms" paper), as seen in Table 5 of the Appendix of the original MixMatch paper.
>
> 4. Another paper, [2], reports very competitive results on CIFAR-10 for 4k labels.
> A: We will include a discussion of this paper in the revised manuscript. Similar to the comment on MixMatch above, we only use small models with 1.5 million parameters compared with the 26 million parameters in SWA. We chose this experimental setting because it simplifies comparison to existing results, as argued in "Realistic Evaluation of Semi-Supervised Learning Algorithms".

---

### Official Review · AnonReviewer3 · 2019-10-24
**Official Blind Review #3**

**Rating:** 6

**Review:**

Summary
The authors make three major contributions that improve MixMatch and achieve state-of-the-art in a semi-supervised image classification task. The major contributions include: (1) distribution alignment to calibrate the predicted distribution of unlabeled data; (2) augmentation anchoring to allow more aggressive data augmentation; and (3) CTAugment to train the augmentation policy alongside the semi-supervised model.
The authors conduct experiments on SVHN, CIFAR-10 and STL, and show significant improvements over the MixMatch baseline. They also show good results (15.08% error rate) of training with 40 labeled data, in spite of very high variation. In the ablation study, they show the error rate drops as K (number of augmentation) increases. They also conduct ablation studies on the design choices of their method.

Decision
The decision for this paper is borderline, tending towards a weak accept. Overall, the paper proposes some simple but interesting ideas, e.g. distribution environments. However, although the proposed method achieves good performance over various (smaller) benchmarks, the method seems ad-hoc and complicated. As pointed out in the weakness section, many design choices are not well motivated, and the effects of those designs are not well studied. The tendency to accept is due to the overall strong results.

Strength
1. Significant improvement over MixMatch baseline.
2. The proposed augmentation anchoring and distribution alignment can be easily integrated into existing work.
3. The proposed CTAugment method lifts the burden of training an RL data augmentation policy.

Weakness
1. The objective of the update equation of CTAugment’s learned weights seems contradicted with the purpose of how data augmentation is used in the consistency-based SSL method. In other words, the objective of the update equation encourages higher weights for the distortion parameter that leads to lower variation in the predicted distribution. However, the idea of aggressive data augmentation is to generate data that has high variation in the model prediction, and then penalize the variation in the form of consistency loss. The variation induced by aggressive augmentation is the root of the consistency loss that helps regularize the model.
2. The authors should provide ablation study and analysis of their CTAugment. For example, they should compare with simple random augmentation policy. It is also recommended to show the learned weights of the distortion parameter. Also does larger K value when applied for vanilla MixMatch approach the results in ReMixMatch?
3. The authors should provide more detail of the setting in the ablation study. For example, the setting of “No strong aug.” and “No weak aug.” are not clear.
4. The authors hypothesize that “stronger augmentation can result in disparate predictions, so their average may not be a meaningful target.” However, they do not show any analysis to support this hypothesis.
5. It is recommended to evaluate the method on larger datasets such as CIFAR-100. It is not clear how well these methods scale, and for example using k=8 adds computation which may hinder training scalability.

Minor Comments
1. For Table 2 and Table 3, it should be “error rate” rather than “accuracy”.
2. How is the loss weight λr tuned in the 40 labeled setting? How are the hyper-parameters tuned in general?


**Experience Assessment:**

I have published in this field for several years.

**Review Assessment: Checking Correctness Of Derivations And Theory:**

N/A

**Review Assessment: Checking Correctness Of Experiments:**

I carefully checked the experiments.

**Review Assessment: Thoroughness In Paper Reading:**

N/A

---

> ### Author Response · Authors · 2019-11-14
> **Official response to blind review #3**
>
> 1. However, although the proposed method achieves good performance over various (smaller) benchmarks, the method seems ad-hoc and complicated. As pointed out in the weakness section, many design choices are not well motivated, and the effects of those designs are not well studied.
> A: While ReMixMatch comprises many components (some of which are new), we believe our ablation study justifies the reason why each component exists. If there are additional ablation experiments that you think would be helpful for us to run, please let us know.
>
> 2. The objective of the update equation of CTAugment’s learned weights seems contradicted with the purpose of how data augmentation is used
> A: It is true that CTAugment at any point in time will only perform augmentations that the model correctly predicts. However, we select augmentations where the probability the model output will change is less than 1. As such, the augmentation boundary will grow progressively as the training process converges. (We experimentally observe this fact: for example, rotation is initially only invariant up to +/- 13 degrees but throughout training becomes invariant to +/- 30 degrees.)
>
> We don’t aim to maximize the output variation at any instant, but instead ensure that by the end of training the model is invariant to large perturbations.
>
> 3. The authors should provide ablation study and analysis of their CTAugment.
> A: As also discussed with reviewer 2, for space reasons we provided only a short description of CTAugment, and how it differs from AutoAugment. We updated the draft to include a longer treatment in the appendix.
>
> 4. The authors should provide more detail of the setting in the ablation study
> A: We agree with the reviewer the details are sparse. We will include more details. To answer the reviewer’s specific questions: “No strong aug.” means that all augmentations were weak (as is done in MixMatch) and “No weak aug.” means that all augmentations were strong. If there are other questions we will clarify any.
>
> 5. The authors hypothesize that “stronger augmentation can result in disparate predictions, so their average may not be a meaningful target.”
> A: See above, where we found that the experiment diverged in the “No weak aug.” ablation (using strong augmentations only).

---

### Official Review · AnonReviewer1 · 2019-10-24
**Official Blind Review #1**

**Rating:** 6

**Review:**

This paper presents ReMixMatch an improved version of MixMatch. The main contributions are the distribution alignment and the augmentation anchoring. Distribution alignment rescales the predictions based on the difference between the model marginals and the ground truth running average estimation. Augmentation anchoring instead of computing the guessed probabilities on unlabelled data as the average probabilities on transformed samples (as in MixMatch), it considers as guessed labels the average probabilities obtained from weak transformations (flip+crop) even when using stronger transformations (Autoaugment like).

The paper is well written, has interesting experiments and very impressive results.
However, there are some negative points that the authors should clarify:
- The final method is a mixup of many different techniques, thus, not a strong contribution, but many smaller contributions.
- As shown in the ablation study, the main contribution on the obtained results seems to be the use of stronger transformations than in MixMatch. This is not so interesting, even though results are impressive. If this is the case, authors should state it more clearly in the paper that a large proportion of the gap in performance between MixMatch and ReMixMatch is the introduction of stronger transformations (Autoaugment style).

Overall the paper is well presented and contributes to further improve the performance on semi-supervised learning. I there fore recommend it for acceptance. However, I would like to see in the paper a more general overview on the fact that strong transformations can further improve semi-supervised methods and ReMixMatch is a way to leverage those transformations.


Additional comments:
- Instead of using the rescaling trick for distribution alignment, what about enforcing the marginal distribution on the annotated data and the marginal distribution of the model to be similar with KL divergence? Would it be better or worse than the proposed approach?

**Experience Assessment:**

I have published one or two papers in this area.

**Review Assessment: Checking Correctness Of Derivations And Theory:**

I assessed the sensibility of the derivations and theory.

**Review Assessment: Checking Correctness Of Experiments:**

I assessed the sensibility of the experiments.

**Review Assessment: Thoroughness In Paper Reading:**

I read the paper thoroughly.

---

> ### Author Response · Authors · 2019-11-14
> **Official response to blind review #1**
>
> 1. The final method is a mixup of many different techniques, thus, not a strong contribution, but many smaller contributions.
> A: While ReMixMatch comprises many components (some of which are new), we believe our ablation study justifies the reason why each component exists. If there are additional ablation experiments that you think would be helpful for us to run, please let us know.
>
> 2. As shown in the ablation study, the main contribution on the obtained results seems to be the use of stronger transformations than in MixMatch.
> A: We actually found that using stronger augmentations in MixMatch resulted in divergence. We mention this in the paper ("Since MixMatch uses a simple flip-and-crop augmentation strategy, we were interested to see if replacing the weak augmentation in MixMatch with AutoAugment would improve performance but found that training would not converge.") but will emphasize this more in the next draft. We also found in our ablation study that using only strong augmentation (i.e., replacing weak augmentations with strong augmentations) resulted in very poor performance for ReMixMatch, suggesting that anchoring towards a weaker augmentation is important. We will update the labels in the ablation table to make this more clear.
>
> 3. What about enforcing the marginal distribution on the annotated data and the marginal distribution of the model to be similar with KL divergence?
> A: We tried this approach in initial experiments and found that it performed poorly. Using the KL loss also introduces a scalar multiplier hyperparameter for this loss term. We spent some time tuning this hyperparameter and were unable to obtain good results, so we chose to use the proposed version which does not introduce such a hyperparameter. It may be that further investigation into this form of a loss could be fruitful.

---

### Decision · Program_Chairs · 2019-12-19

**Decision:**

Accept (Poster)

**Comment:**

This works improves the MixMatch semi-supervised algorithm along the two directions of distribution alignment and augmentation anchoring, which together make the approach more data-efficient than prior work.
All reviewers agree that the impressive empirical results in the paper are its main strength, but express concern that the method is overly complicated and hacking together many known pieces, as well as doubt as to the extent of the contribution of the augmentation method itself, with requests for better augmentation controls.
While some of these concerns have not been addressed by authors in their response, the strength of empirical results seems enough to justify an acceptance recommendation.